# Effects of Power and Laser Speed on the Mechanical Properties of AlSi7Mg0.6 Manufactured by Laser Powder Bed Fusion

**DOI:** 10.3390/ma15238640

**Published:** 2022-12-04

**Authors:** Sébastien Vaudreuil, Salah-Eddine Bencaid, Hamid Reza Vanaei, Anouar El Magri

**Affiliations:** 1Euromed Polytechnic School, Euromed Research Center, Euromed University of Fes, Route de Meknès (Rond Point Bensouda), Fès 30 000, Morocco; 2Research Center, Léonard de Vinci Pôle Universitaire, 92916 Paris La Défense, France; 3Arts et Métiers Institute of Technology, CNAM, LIFSE, HESAM University, 75013 Paris, France

**Keywords:** laser based powder bed fusion, LBPF, selective laser melting, SLM, aluminum, AlSi7Mg0.6, mechanical properties, Stripes, Chess, heat treatment, laser parameters

## Abstract

The AlSi7Mg0.6 alloy, with its good tolerance against strain, is used in laser powder bed fusion (LPBF) to produce parts with complex geometries for aerospace engineering. Production of parts with good mechanical strength requires, however, the optimization of laser parameters. This study thus evaluated the influence of scanning speed, laser power, and strategy on several mechanical properties (tensile/resilience/hardness) to identify an optimal processing region. Results have shown the profound influence of laser power and scanning speed on mechanical properties, with a limited influence from the laser strategy. Tensile strength values ranging from 122 to 394 MPa were obtained, while Young’s Modulus varied from 17 to 29 GPa, and the elongation at break ranged from 2.1 to 9.8%. Surface plots of each property against laser power and speed revealed a region of higher mechanical properties. This region is found when using 50 µm thick layers for energy densities between 25 and 35 J/mm^3^. Operating at higher values of energy density yielded sub-optimal properties, while a lower energy density resulted in poor performances. Results have shown that any optimization strategy must not only account for the volumic energy density value, but also for laser power itself when thick layers are used for fabrication. This was shown through parts exhibiting reduced mechanical performances that were produced within the optimal energy density range, but at low laser power. By combining mid-speed and power within the optimal region, parts with good microstructure and limited defects such as balling, keyhole pores, and hot cracking will be produced. Heat-treating AlSi7Mg0.6 parts to T6 temper negatively affected mechanical performances. Adapted tempering conditions are thus required if improvements are sought through tempering.

## 1. Introduction

Additive manufacturing (AM) represents a real breakthrough for the automotive, biomedical, and aerospace sectors. A major strength of this technique is its capability of producing complex shapes, giving more freedom to engineers and designers to innovate [1,2]. The laser powder bed fusion (LPBF), also commonly called selective laser melting (SLM), is the most mature process for manufacturing a fully dense product with good strength from thin layers of metal powder [3]. Combined with the use of lightweight metals, significant weight reductions are achievable while maintaining adequate mechanical properties. 

Aluminum alloys have shown great potential because of their high strength-to-weight ratio, corrosion resistance, and good mechanical properties. While a wide choice of aluminum alloys is available for foundry, extrusion, or cold-work applications, the choice of alloys commercially available for LPBF is limited [2,3,4,5]. The most readily available alloys in the Al–Si and Al–Si–Mg families are AlSi10Mg and AlSi12, as they are the oldest aluminum alloys used in LPBF. The use of silicon in these alloys improves fluidity and fusion, while magnesium leads to increased strength through solid solution strengthening [4]. These benefits explain the growing uses found for these alloys in LPBF and the number of publications aimed at better understanding the influence of processing parameters [6,7,8,9,10,11,12,13,14,15,16,17,18]. Fewer investigations have been conducted on AlSi7Mg0.6 [19,20,21,22,23,24,25,26,27,28,29] because of its late 2015/early 2016 availability as LPBF-grade powder, compared to 2011 for AlSi10Mg. This alloy has the advantage of being more insensitive to hot cracking, making it interesting for automotive and aerospace applications. 

Several investigations on AlSi10Mg and AlSi7Mg0.6 using LPBF have been aimed at improving performances while reducing process variability. Pereira et al. [26] thus classified the areas of primary research focus into four categories, covering the optimization of process parameters, the influence of build orientation, the development of heat treatment strategies, and the numerical analysis of LPBF parts performance. While LPBF processing of AlSi10Mg and AlSi12 is better understood thanks to a large number of studies, the increasing number of studies on AlSi7Mg0.6 are now giving more insight for this alloy. Many of these studies evaluated the influence of laser parameters, build orientation, or in situ/ex situ heat treatment on part quality. However, this can be very demanding, as many performance criteria are available such as mechanical properties, density, porosity, residual stresses, compositional change, or microstructure. Similarly, Zhang et al. [5] and Rometsch et al. [16] reported the same approach to optimize SLM-printed aluminum components.

These criteria are often related to one another, as higher part density generally leads to higher mechanical properties. Rao et al. [19] evaluated the influence of laser-related parameters, using the relative density to identify optimum operating conditions for AlSi7Mg0.6. A maximum density of 99.8% was achieved at 50 J/mm^3^, with increasingly higher porosity observed the further energy is reduced. This could be attributed to an incomplete part consolidation and insufficient powder melting occurring at lower energy density that promote pores formation. At optimum laser parameters, ultimate tensile strength (UTS) between 290 and 426 MPa was achieved, with the best properties obtained for horizontally built samples at low platen temperature [19]. A similar study by Pereira et al. [26] yielded densification greater than 99.95% in the 70–85 J/mm^3^ energy density range, resulting in UTS between 435 and 446 MPa. These energy density values for AlSi7Mg0.6 are in line with the 52 J/mm^3^ and 75–94 J/mm^3^, respectively, found by Wei et al. [13] and Zhao et al. [30] for AlSi10Mg. These results were obtained for builds performed at high energy density, something achieved by using 30 µm thin layers of powder. While this helps achieve a high energy density, these thin layers can be detrimental to productivity, as it requires up to 67% more layers to achieve the same build height, thus increasing build duration. Thicker layers will be preferred by industrial users to achieve increased productivity if mechanical properties can be maintained. However, for LPBF systems with maxed-out laser power, the increased layer thickness will reduce the applied energy density, possibly leading to a reduction in properties.

In addition to laser parameters, the mechanical properties of as-built parts are also susceptible to the platen temperature [11,19,25] as well as the part orientation [19,24] used during LPBF. In the case of platen temperature, Buchbinder et al. [11] observed that heating the building platform to 220 °C during the LPBF process of AlSi10Mg caused the solidification of a different microstructure, lowering its mechanical properties. Rao et al. [19] also observed changes in microstructures between samples produced at platen temperature of 35 and 200 °C, with the latter showing signs of over-aging leading to poorer mechanical properties. Part orientation also plays a crucial role by orienting the microstructure formed parallel or perpendicularly to the strain direction. It was observed by [19,24] that tensile bars made horizontally exhibited higher tensile strain and strength values than bars made vertically. Oliveira de Menezes [24] studied the mechanisms of fracture propagation in samples built in various orientations. They observed lower mechanical properties when a fracture propagated perpendicularly to the building direction, as it moved at the interface between the melt pools of two successive layers. This occured in tensile samples built vertically. For fracture propagating along the building direction, which happened in horizontally built tensile bars, the fracture moved through the solidified melt pools of all layers without any specific path.

Rao et al. [20] investigated the links between microstructures and tensile properties in the as-built and heat-treated states. They found that the higher tensile properties of printed A357 samples compared to their casted counterparts can be attributed to the ultrafine microstructure composed of eutectic nano-sized Si particles found in as-built LPBF parts, which result in higher strength but lower ductility than cast parts [19,20]. The formation of this microstructure is affected by the cooling rate through the built platform temperature. Aversa et al. [21] investigated the effects of build platform temperature on mechanical performances for AlSi7Mg0.6. Using 30 µm layers, they produced parts using a laser power of 195 W, 1200 mm/s scanning speed, 0.1 mm hatch distance, and Stripes scanning strategy. The highest mechanical properties were achieved when the platform temperature was set at either 140 or 170 °C, yielding a maximum tensile strength of 408 MPa and yield strength of 284 MPa. They observed a low yield strength when the platform temperature was above or below this range. At low platform temperatures, the resulting high rate of cooling favored a quenching effect that led to an under-aging of the alloy. The lower silicon precipitate content formed decreased mechanical properties. At the other end, a too-high platform temperature will over-age the alloy by promoting Si diffusion into stables precipitates.

It is well known that LPBF produces parts exhibiting pronounced columnar morphology, with significant differences between in-plane and build direction morphology [31]. The use of suitable heat treatment could lead to better mechanical performances, but the microstructure and properties of A357 samples produced by LPBF are still being studied to find the right process parameters. Trevisan et al. [32], for example, investigated the impact of the post-processing heat treatment on the mechanical properties of A357 LPBF samples and found that tensile strength decreased after stress relief heat treatment as well as with T6 heat treatment. This decrease was attributed to the columnar microstructure becoming more isotropic with the increasing harshness of heat treatment. Likewise, Yang et al. [31] presented a study on the effect of heat treatment on the mechanical properties and microstructure of A357 after stress relief. For as-built A357 SLM samples, the rapid melting and cooling cycles affected the intermetallic phases such as Mg2Si precipitates. The heat treatment process caused a breaking up of the Si network, leading to high ductility, while Si particles at grain boundaries were observed to coarsen. However, as the microstructure became more homogenized, the anisotropy progressively disappeared and affected yield strength and ductility. It is therefore important to understand the effects of the post-processing heat treatment on LPBF parts [33]. 

The present work aims to evaluate the impacts of energy density and laser strategy on the mechanical properties of the AlSi7Mg0.6 alloy when thick layers are used. An emphasis will be placed on finding optimal manufacturing conditions for a given laser strategy, relying on the tensile, impact and hardness properties. These results will be compared to those from works performed at a lower layer thickness to evaluate if comparable mechanical properties can be achieved using thicker layers. This could help achieve higher build productivity through a significant reduction in the number of layers required, thus lowering the total build time. However, that would be only if mechanical properties can be maintained. The effects of T6 heat treatment on mechanical properties will also be evaluated.

## 2. Materials and Methods

The AlSi7Mg0.6 powder used in this study was produced by argon gas atomization (LPW Technology). This atomization process resulted in spherically shaped particles with a size distribution lesser than 63 µm in diameter. Before any use of powder, sieving through a 100 µm sieve was performed to remove any large particles that could be present. All fabrications were performed under a nitrogen atmosphere on an SLM 125HL (SLM Solutions GmbH, Lübeck, Germany) equipped with a 400 W fiber laser operating at 1075 nm wavelength (IPG Photonics), resulting in a laser spot size of 100 µm. A layer thickness of 50 µm, hatch distance of 170 µm, and platform temperature of 120 °C were used in all fabrications. These values are based on SLM Solutions supplied parameters for this alloy, albeit SLM Solutions recommends a platform temperature of 150 °C and the Stripes strategy. A lower platen temperature than recommended was used here to test its effects on mechanical properties. Only for fabrications performed using SLM recommended parameters that 120 and 150 °C platen temperatures were used for fabrication.

A centered composite design experimental plan was used for both the Stripes and Chess laser strategies (Figure 1) to investigate the effects of laser power and speed on mechanical properties. Five levels of laser power and five laser speeds were thus evaluated for each strategy, using a fixed hatch distance of 0.17 mm. This yielded nine levels of volumic energy density, ranging from 12.75 to 69.11 J/mm^3^, calculated using Equation (1) below: (1)$$E_{v}=\frac{P}{v\cdot h_{D}\cdot e_{C}}$$
where *P* is the laser power (W), v is the laser speed (mm/s), $h_{D}$ is the hatch distance, and $e_{C}$ is the layer thickness (mm). These parameters, listed in Table 1, also include the recommended parameters by SLM Solutions for this specific alloy, as well as the linear energy density calculated with Formula (2):(2)$$E_{l}=\frac{P}{v}$$

Three types of testing were performed to characterize mechanical properties, namely tensile, resilience (Charpy), and hardness (Rockwell type B). Each build platen produced with either one of the 19 sets of parameters thus contained 12 tensile test bars of the ASTM E8M Type 3 design, as well as 8 ASTM E23 impact test bars. All 20 samples per platen were vertically printed, with a minimum distance of 7 mm between each part to limit spatter contamination and prevent part overheating. Figure 2 shows their placement on the platen and a typical de-powdered platen after fabrication in one of the conditions listed in Table 1. Hardness measurements were performed on the impact test bars. 

Tensile testing was carried out on an MTS Criterion Model 45, using a traverse speed of 0.54 mm/min to achieve the desired load rate. A 100 kN load cell, with a precision of 0.5% of the read value, recorded continuously the applied load. Of the twelve E8M tensile bars produced in each series, six were randomly selected for testing in the “as-produced” state, i.e., without any post-processing operations (heat treatment or milling to smooth surface), to obtain a minimum of four similar results. Resilience testing was performed on three randomly selected E23 test bars for each series, using a JBS-500 impact tester (Chengyu Testing Equipment Co., Ltd., Qingdao, China) equipped with a 250J pendulum. Before testing, a 2 mm deep V-shaped notch was made in the sample by a broaching machine. Hardness was measured with an HRS-150 Digital display Rockwell tester (LaiZhou Weiyi Experimental Machinery Manufacture Co., Ltd., Laizhou City, China), using the samples prepared for impact testing. The Rockwell B scale (HRB) was used, as it is more suitable for aluminum alloys. The 1.588 mm steel ball indenter applied a 60 kg load to the sample, resulting in a 981 N total force. Hardness measurements were performed on three different locations of the sample to obtain an average value.

In addition to mechanical testing, the density of the SLM samples was measured using Archimedes Method. This density was calculated using Equation (3) [19]: (3)$$\rho_{s}=\frac{M_{A}*\rho_{L}}{M_{A}-M_{w}}$$
where $\rho_{L}$ is the density of the liquid used (water in this case), $M_{A}$ is the mass of the sample in air, and $M_{w}$ is the mass of the sample fully submerged in water. Three measurements were carried out to average results.

According to the literature, the theoretical density of A357 aluminum alloy is 2.68 g/cm^3^. The relative density $\left(\rho_{R}\right)$ in % is thus obtained by dividing the value of the measured density $\left(\rho_{s}\right)$ by the theoretical value $\left(\rho_{\mathrm{Th}}\right)$, following Equation (4):(4)$$\rho_{R}\left(\%\right)=\frac{\rho_{s}}{\rho_{\mathrm{Th}}}\times 100$$

The microstructure of the fracture surface for three tensile bars was evaluated by scanning electron microscopy (SEM), using a Quanta 200 ESEM (Thermo FEI, Eindhoven, the Netherlands). Three samples were selected from different series to evaluate the effect of energy density and heat treatment. Two samples were from the series exhibiting the best and the worst mechanical properties, while the third sample was the T6 heat-treated version of the best series.

The effects of heat treatment on mechanical properties were also evaluated. Being commonly used for aluminum casted parts, the T6 temper was selected. This involves a solution treatment carried out at 540 °C for 3 h, followed by quenching in 80 °C water before a 3 h aging step at 155 °C to achieve precipitation hardening through Mg2Si formation. The remaining six E8M tensile bars and three E23 test bars were treated to T6 temper.

## 3. Results

Many authors use relative density as a simple screening way to evaluate the impacts of parameters on part quality [19,26]. Figure 3 shows the surface plot for the relative density of as-built samples produced with both laser strategies evaluated. In the case of Stripes (Figure 3A), the optimal region is located between 225 W and 350 W for laser power and 800 mm/s and 1400 mm/s for laser speed. Low relative densities are found in samples produced at low laser power and high scanning speed, two conditions giving rise to melting pool instability that leads to decreased density of the material. It is even possible in these conditions that particles are not completely melted, resulting in pore formation between layers. The maximum relative density, at ~98.2%, was attained for samples built at an energy density between 25.9 and 27.5 J/mm^3^. This energy density results from either a high laser power/speed or a middle laser power/speed. This maximum density of 98.2% is higher than the 96.4% density obtained by Kan et al. [17] for AlSi10Mg processed at 44.3 J/mm^3^ (50 µm layer, 350 W, 930 mm/s, and 170 µm hatch distance). Test conditions #9 used parameters close to those of Kan et al. [17], yielding a higher density of 97.9% than Kan et al. [17]. This shows the susceptibility on final properties of the multiple process parameters, as shown by Giovagnoli et al. [18]. The highest relative densities have been reached with a laser power above 225 W combined with a moderately high speed. Rao et al. [19] reported a relative density of 99.79% achieved at 300 W laser power and 2000 mm/s in laser speed, which is 1.5% higher than the 98.22% density achieved here for samples produced at 315 W and 1430 mm/s (Test conditions #8). The use of tighter hatching (100 µm instead of 170 µm) and thinner layers (30 µm instead of 50 µm) by [19,26] could explain the higher densities achieved. Aboulkhair et al. [8] have shown that reducing the hatching distance from 150 µm to 100 µm can increase density by up to 1.5%, especially when a low power setting is used. This is because more heat can accumulate in the melt pool, resulting in a more homogeneous layer because of the slower cooling. Thinner layers may also be more susceptible to the depth reached by the melt pool, something that could affect bonding between neighboring layers. Experimental work coupled with simulation by Li and Gu [7] has shown melt pool depths exceeding 100 µm when high power is combined with low scanning speed. Similar work by Mauduit et al. [29], in which they calculated melt pool dimensions of AlSi7Mg0.6 as a function of the linear energy density, showed melt pool depths exceeding 400 µm for linear energy density above 1 J/mm. The linear energy densities normally found in most works, which are between 0.15 J/mm and 0.30 J/mm, would yield melt pool depths varying between 24 and 125 µm based on Mauduit et al. [29]. This implies that the melt pool can extend, depending on process parameters, beyond the first underlayer. While a deeper molten pool may improve bonding between layers, this may also favor porosity formation, especially if the hatch distance is large. This can be seen when comparing results from this work to those of Rao et al. [19] on the basis of porosity against linear and volumic energy densities (LED and VED). While a 99.79% relative density was obtained by Rao et al. [19] for an LED of 0.15 J/mm, VED of 50 J/mm^3^, and hatch of 100 µm, the settings used in this work resulted in a 98.2% relative density at higher LED of 0.23 J/mm, VED of 25.9 J/mm^3^, and 170 µm hatch distance. The expected melt pool depth would be ~24 µm for Rao et al, which is less than their 30 µm layer, while the expected melt pool depth of ~77 µm in this work is greater than the 50 µm layer. Pereira et al. [26] achieved an even higher density at 99.96% by operating at higher LED and VED (0.27 J/mm and 69.0 J/mm^3^, respectively). While their melt pool would be much deeper than the 30 µm layers used, Pereira et al. relied on a hatch distance of 130 µm to better distribute the applied heat and reduce the formation of porosity.

Another process parameter that could possibly influence porosity is the choice of laser strategy. While many commercial LPBF systems rely on the Stripes strategy, some systems rather operate following a Chess pattern. Looking at the surface plot of the Chess strategy (Figure 3B), a similarity to the Stripes plot can be observed, including the optimal region. A maximum relative density of 98.50% was achieved with the Chess strategy, a level similar to the 98.2% obtained using Stripes. This indicates that the laser strategy (Stripes versus Chess) does not affect the resulting relative density. It will affect, however, the build time, as a greater number of shorter paths and more direction changes are done in the Chess mode. 

A strong link was shown between relative density and mechanical properties, where the densest part tended to exhibit the highest mechanical properties [19,26]. This was observed when comparing the surface plot of relative density (Figure 3) to the surface plot of the tensile strength (Figure 4). Both surface plots display the same overall appearance and optimal regions. Figure 4A shows the surface plot for tensile strength obtained for samples produced with the Stripes laser strategy. No significant necking was observed in any of the samples tested. The highest tensile strength reached was between 380 to 400 MPa, and was achieved with power and speed ranging, respectively, from 225 W to 300 W and 950 to 1400 mm/s. The energy density for this optimal region was between 25 to 28 J/mm^3^. By comparison, a volumic energy density of 35.8 J/mm^3^ resulted from the laser parameters supplied by SLM Solutions for AlSi7Mg0.6. These parameters resulted in a tensile strength of 386.6 MPa when the platform temperature was 120 °C, and 407.0 MPa at 150 °C. This increase in tensile strength with platform temperature is in line with observations made by Aversa et al. [21].

Some power and speed combinations used in this work resulted in tensile strength as low as 140 MPa for AlSi7Mg0.6, something occurring at insufficient laser power or too fast scanning speed. As these conditions favor melt pool instabilities or incomplete particle melting, this will increase the formation of pores between layers that negatively affect mechanical properties. Good tensile strength properties are thus achieved through sufficient laser power and moderately high speed, something consistent with the results ranging from 380 to 395 MPa measured by Rao et al. [19], Aversa et al. [21], and Zvoníček et al. [27] in AlSi7Mg0.6 alloy. It must be noted that these authors used a smaller layer thickness (30 µm instead of 50 µm) and tighter hatching (100 µm instead of 170 µm). This helped them achieve energy densities between 50 and 58 J/mm^3^, which is twice the energy density used in this work. Such a large difference in energy density could explain the lower relative density obtained in this work compared to Rao et al. [19] and Pereira et al. [26]. 

Other tensile properties also show the influence of laser power and scanning speed. Figure 4C,E shows that both Young’s Modulus and elongation at break exhibit an optimal region, albeit with different values of laser power and speed. In the case of Young’s Modulus (Figure 4C), the optimal region yielded a maximum value of 28.88 GPa at low power and speed (155 W, 580 mm/s), yielding an energy density of 31 J/mm^3^. This Young’s Modulus value is close to the 32 GPa found by Zyguła and al. [14] for AlSi10Mg parts produced at 24 J/mm^3^ with layers of 50 µm. By comparison, the use of the laser parameters supplied by SLM Solutions for AlSi7Mg0.6 yielded a Young’s Modulus of 22.5 GPa at 35.8 J/mm^3^ when the platform temperature was fixed at 120 °C, and of 28.6 GPa at 150 °C. The Modulus values achieved with 50 µm layers are, however, much lower than the 62 GPa value reported by Pereira et al. [26] for 30 µm AlSi7Mg0.6 layers. Such a difference in behavior might find its origins in the extra remelting of the interfacial region between layers that occurs when thin layers are used. While the same amount of thermal energy is applied for both thicknesses, based on a ~0.27 J/mm linear energy density used, the use of thin layers enables a deeper diffusion of this thermal energy into extra layers. This can help achieve more favorable internal metallography through the formation of silicon precipitate, thus improving Young’s Modulus. Using a combination of low power (155 W and less) and high speed (1000 mm/s and more) for 50 µm layers will result in Young’s Modulus values as low as 16.95 GPa. Interestingly, Young’s Modulus shows a more pronounced effect on changes in scanning speed from the optimum region than changes in laser power. Doubling the laser power at low scanning speed will reduce by 8% the measured Young’s Modulus, while doubling scanning speed at low power will almost halve it. Such negative impact on ductility from the scanning speed at low speed could be attributed to a combination of reduced melt pool depth at low LED that decreases interlayer adhesion, and the lesser time where the alloy is in liquid phase, resulting in less Si precipitates and dendrite growth. For any scanning speed used, the effects of laser power on Young’s Modulus become less prominent when sufficiently high laser power is applied. 

The deformation at break for parts produced at mid-laser power and speed was found to be between 4.5 and 8.4% (Figure 4E). This is similar to or slightly higher than the 3 to 5.2% deformation at break measured for samples produced at twice the energy density, but with 30 µm layers [19,21,26]. At high laser power and speed, the deformation at break reached 9.80% at half the energy density used by Zvoníček et al. [27], who attained 9.85% using 50 µm layers. This is similar to the 10.1% deformation at break obtained when using the SLM-supplied parameters for AlSi7Mg0.6 with a platform temperature of 120 °C, and slightly higher than the 8.6% obtained at 150 °C. This decrease in elongation at break with increased platform temperature is coherent with observations made by Aversa et al. [21]. Dissimilar microstructures could explain such variability in the deformation at break, especially since samples produced with a platform temperature of 120 °C are in a metastable condition consisting of supersaturated solid solution [21]. As process parameters vary, this will affect microstructures formed during cooling that may or may not enhance the sample deformation before break, even for the same energy density. 

Resilience and hardness are also affected by power and speed (Figure 5), with the middle level of power and speed yielding the best values at, respectively, 1.01 J/cm^2^ and HRB 85 (roughly equivalent to 160 HB10 and 160 HV). A lower resilience was achieved compared to the 2.5 J/cm^2^ obtained by Raus et al. [34], while a greater hardness was achieved compared to the values of 98 HB10 obtained by Tonelli et al. [28], 143 HV 0.3 by Mauduit et al. [29], and 139 to 149 HV by Raus et al. [34]. These changes could be explained by the lower volumic energy density resulting from the thicker layers used (50 µm against 20 and 30µm), while a similar linear energy density was used (between 0.21 and 0.24 J/mm). This could result in finer microstructures, thus increasing hardness but also brittleness. The fine microstructure generally observed in SLM-produced material leads to higher hardness values than those of cast material. As shown in Figure 4, each property exhibits a different optimal region of laser power and scanning speed. Any optimization strategy of laser parameters will thus imply a tradeoff between various mechanical properties.

To better understand the changes in properties resulting from laser parameters, it is important to evaluate any changes in the microstructure of SLM-produced parts. It is well known that the metallurgy of SLM-built parts differs from that of conventionally produced parts, thus yielding distinctive mechanical properties [26]. The use of a localized moving heat source in the SLM process leads to significant microstructural heterogeneity during part formation. The resulting microstructure, consisting of supersaturated Al along with Si-rich areas, will exhibit a mixture of columnar and isotropic grains with a periodicity corresponding to layer height, line width, and scanning pattern used [31]. These crystallographic heterogeneities are greatly influenced by laser power and scanning speed. To better understand how these two parameters affect mechanical properties, the fracture surfaces of tensile bars were observed by scanning electron microscopy. Figure 6 shows the fracture surface of as-built samples exhibiting the highest (Test 5) and lowest (Test 3) tensile strength. Test 5 was produced in the optimal region for tensile strength, at 235 W and 1005 mm/s, while Test 3 used a low laser power and high speed (155 W and 1430 mm/s). While the fracture surface for both tests showed an instantaneous rupture, a difference in microstructure can be observed. SEM revealed a flat and brilliant rupture surface for Test 5, with cleavage planes exhibiting no distinguishable direction of propagation (Figure 6a). In the case of Test 3, an oriented fracture surface with numerous cups and inclusions was observed (Figure 6b). The observed differences in rupture surface could be associated with changes in the general structure of the molten region from the lower laser power. The effects of laser conditions on the alloy’s morphology and microstructure are better highlighted at higher magnifications. Figure 6c,d obtained at a magnification of 600X, shows the presence of balling. This phenomenon occurs when the molten track narrows and decomposes into a row of spheres to reduce surface tension when molten material is not able to wet the underlying substrate. This balling effect, which is more extended at low laser power (Test 3), could be attributed to an insufficient supply of energy that results in the incomplete fusion of powder particles. The hot cracks observed for Test 3 result from an insufficient amount of energy supplied by the laser. In this case, the laser is unable to create a liquid metal pool large enough to properly fill spaces between hatches. Such hot cracking can be detrimental to mechanical properties if this phenomenon is extensive throughout the part.

It is well-known that the as-built SLM microstructures tend to exhibit a fine cellular-dendritic structure. In the case of aluminum alloys, the Al dendrite cells are enveloped by a thin eutectic layer that restricts the dislocation motion, thus characterizing the properties of SLM-produced parts. To achieve a better understanding of these interactions, Romanova et al. [35] studied numerically the grain shape and texture effects in terms of micromechanical simulations. They observed the formation of two distinct patterns within each melt pool, where the central region comprises a radial pattern of elongated columnar grains and fine equiaxed grains are found at the boundary. In their studies, they evaluated models with random and cube-textured columnar grains on the stress–strain distributions inside the melt pools and adjacent regions, and found that cube-textured columnar grains are more homogeneous, thus reducing high stress concentration in them.

Yan et al. [36] developed an integrated dendrite and eutectic evolution model to predict the Al–Si eutectic evolution and morphologies. From their simulated results, they showed a refining of dendrites with increasing cooling rates and a sensitivity to eutectic undercooling where higher undercooling refined the eutectic microstructures. 

Another parameter that may affect mechanical properties is the laser strategy. Using the same levels of power and speed conditions in each strategy, the Stripes and Chess models were compared. Figure 4A,B shows the tensile strength of as-built samples for both strategies. It can be observed for both strategies that power and speed influence the resulting tensile strength. The main difference is that, while both strategies achieve near identical maximum tensile strength, a larger region of maximum strength is attained through the Stripes strategy. This larger region offers a broader choice of power and speed to select as optimum operating parameters. This could be attributed to the prolonged “close-packing” of laser pathways that maintain a “hot zone” around the melt pool, thus helping microstructure formation in less energetic conditions. Similar observations were found in the cases of Young’s Modulus and deformation at break (not shown). This overall similarity shows that laser strategies have a limited impact on mechanical properties, especially when compared to the individual effects of laser power and scanning speed. The strategy choice will affect more heat evacuation throughout the part and powder bed during fabrication, especially when supports are used, than mechanical properties. This can be explained by the fact that the extra laser pathways generated in the Chess strategy increase the time allowed to dissipate thermal energy in the underlying metallic layers compared to the Stripes strategy. Heat evacuation was not evaluated in the current study, as parts were built directly on the platen to improve the evacuation of thermal energy.

While Figure 4 and Figure 5 give insights into the specifics of laser power and speed on mechanical properties, the volumic energy density (EV) is commonly used to describe the overall state of build parameters. This approach can be convenient because it reduces to a single indicator the four major contributors to melt pool dynamics. The mechanical properties were thus plotted as a function of energy density, ranging from 12.7 J/mm^3^ to 69.11 J/mm^3^, to identify optimum operating regions. Figure 7 illustrates the tensile properties of AlSi7Mg0.6 as a function of energy density for builds realized using the Stripes laser strategy, while Figure 8 illustrates the changes in resilience and hardness. Similar results were obtained using the Chess strategy, confirming the limited impact of laser strategy on mechanical properties. 

It can be observed from Figure 7 that very poor tensile properties are obtained at low energy density where low power combined with high speed was used. These poor properties are the results of internal defects caused by the lack of fusion from an insufficient energy transfer to the powder bed. On the other end, using very high energy densities results in intermediate mechanical properties. This can be attributed to the fact that melt pool dynamics are affected by excessive energy, either from too much laser power or a too-slow speed. When too much laser power is used, the higher melt pool temperatures attained increase the internal recoil pressure generated. This was shown to increase both spattering and keyhole pore formation, both phenomena detrimental to mechanical properties because of the local defects they create. The melt pool will be affected differently when using a too-low scanning speed. The melt pool becomes, in this case, larger as the energy source stays longer at the same place. In addition to localized overheating, the lateral and vertical heat diffusion that occurs while the laser moves away will affect in various ways the microstructure of neighboring laser tracks.

In between these two extremes, an energy density in the 25 to 30 J/mm^3^ range was found to yield the best properties. This is about half the energy density used by others [19,21,26,27] to achieve the same tensile strength of 380 to 395 MPa for AlSi7Mg0.6, while the 52 J/mm^3^ used by Zhao et al. [30] for AlSi10Mg resulted in a 241 MPa tensile strength. While a difference in alloy could explain some variation in optimum energy density, the results obtained show that the optimal zone for AlSi7Mg0.6 seems to be in the 25–35 J/mm^3^ range. Using the parameters supplied by SLM Solutions for AlSi7Mg0.6, which correspond to an energy density of 35.8 J/mm^3^, a tensile strength of 386 MPa was obtained with a build platen set at 120 °C. The lower mechanical properties achieved at 31 J/mm^3^ could be explained by the low laser power used (155 W) that favors hot crack formation through a small melt pool formation. This reinforces the need to use sufficient laser power to achieve a large enough melt pool. Plotting the mechanical properties as a function of energy density helped establishing regression equations for each property considered (Table 2). Because the energy density affects differently each mechanical property, these equations can help in optimizing laser parameters to achieve a compromise for the as-built parts.

As mentioned earlier, the metallurgy of SLM parts differs from parts produced by casting, for example. Because casted parts often experience heat treatment to improve their mechanical properties, it is thus of interest to see if similar results can be achieved for SLM-produced parts. To this effect, the commonly used T6 temper was chosen, where improved mechanical properties are achieved through precipitation hardening. T6 temper implies solution treating the part at high temperatures, followed by artificial aging at low temperatures. For all series, half of the samples were heat treated to T6 temper before performing tensile, impact, and hardness testing. It was observed that some heat-treated samples exhibited blisters, without any specific reasons identified for this phenomenon, and were thus rejected.

Results showed a general decrease in the samples’ tensile strengths following heat treatment throughout the range of power and speed tested (Figure 9). Samples produced at optimum power and speed exhibited a reduction of 40% in maximum tensile strength after heat treatment, from 393 MPa in the as-built state down to 233 MPa. Similar behavior was observed, albeit to a lower extent, for the Young’s Modulus, while elongation at break rather showed an increase after heat treatment. This is in general agreement with works by Aboulkhair et al. [37], where similar behaviors in tensile properties were observed following heat treatment. This negative contribution of heat treatment to mechanical properties can be linked to unfavorable changes in the alloy’s microstructure. This includes the replacement of the initial mixture of small columnar and isotropic grains by more isotropic grains and the larger presence of dispersed Si dendrites that have grown uniformly in the Al matrix during aging. Kan et al. [17] thus observed that the T6 treatment caused a dissolution of the nano-crystalline Si, which then re-precipitate into ~4 μm crystals, thus lowering hardness and tensile properties. Maamoun et al. [38] have highlighted the microstructure evolution for AlSi10Mg as a function of heat treatment temperature and time, where the inhomogeneous fine-grained microstructure progressively becomes a coarse-grained microstructure with increasing temperature. Aversa et al. [21] observed that properties of under-aged samples, e.g., produced using a platform temperature of 100 °C, benefit from a short duration heat treatment at 170 °C through precipitation of Si from a metastable state. In the case of over-aged samples, e.g., produced using a high platform temperature, the same heat treatment only results in a decrease in properties. Per Aboulkhair et al. [9,38], heat treatment conditions must be adapted to SLM-produced parts to account for their unique as-built microstructures if improved mechanical properties are sought.

## 4. Conclusions

This study on AlSi7Mg0.6 aluminum alloy evaluated the influence of laser power, scanning speed, and strategy on several mechanical properties, to identify the optimal processing region when thick layers are used. Being considered a hardenable alloy, the influence on mechanical performances of T6 heat treatment on AlSi7Mg0.6 was also evaluated. 

Results from this study confirmed the profound influence of laser power and scanning speed on mechanical properties when thick layers are used, with the best results obtained at middle power and speed. Such operation at mid-range volumic energy density reduced the balling phenomenon normally associated with incomplete powder melting and yielded a better microstructure. Testing at various power and speed settings also showed that using only volumic energy density as a defining criterion is not enough, especially for thick layers. Any optimization strategy must also consider the laser power itself, as parts with reduced mechanical performances were produced within the optimal energy density range but at low applied power. In such conditions, the smaller melt pool formed during lasing increases the risks of hot cracking during solidification. It was observed that reducing the energy density through reduced laser power will further worsen performance degradations, with parts exhibiting extensive balling and poor microstructure. On the other hand, using a too-high energy density was shown to be detrimental to mechanical properties, as the stronger recoil pressure generated by the laser will increase spattering and keyhole pore formation. High mechanical properties and favorable microstructure will thus be achieved using laser and speed parameters in the optimal region to avoid high porosity and limit the balling phenomenon. From this study, the laser strategy was found to have limited influence on mechanical performances, with both the Stripes and Chess strategy yielding similar values. Using the Chess strategy will, however, lead to longer build time, as a greater number of shorter paths and more direction changes are done, all of which slow down the overall process.

Heat-treating SLM-produced AlSi7Mg0.6 parts showed that applying the standard tempering conditions used for casted parts will have a mostly negative influence on tensile properties. Only for some applications could the increase in elongation observed after heat treatment be beneficial. Adapted aging or tempering conditions must thus be applied if improvements are sought through some form of tempering. 

While energy density remains an important criterion in selecting operating conditions, this study shows it is not an absolute factor to achieve the best mechanical performances. Laser power plays a significant role in the final morphology through its influence on the melt pool. A compromise between laser power and scanning speed to achieve the identified optimal energy density is necessary to achieve parts with high mechanical properties.

## Figures and Tables

**Figure 1 materials-15-08640-f001:**
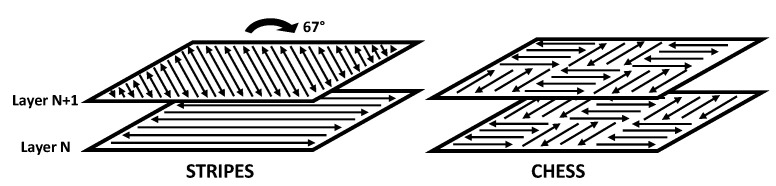
The two laser strategies used in this study.

**Figure 2 materials-15-08640-f002:**
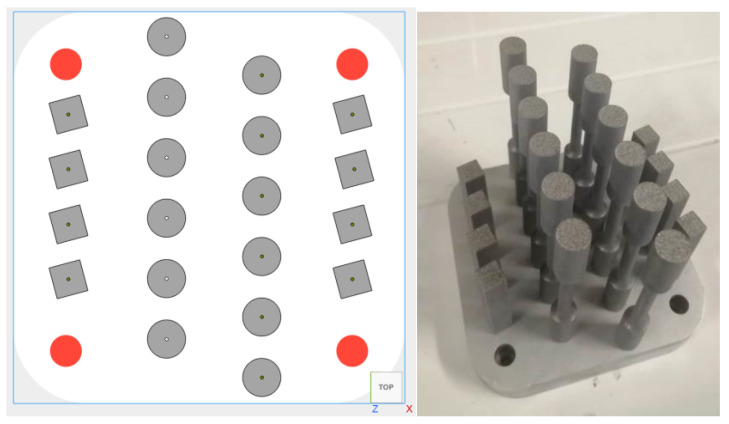
Parts placement on the build platen and a typical platen after de-powdering.

**Figure 3 materials-15-08640-f003:**
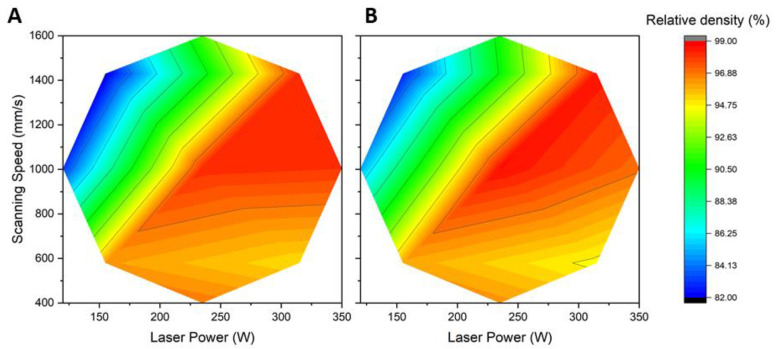
Relative density of “as-built” samples as a function of power and speed for the two laser strategies tested: (**A**) Stripes (**B**) Chess.

**Figure 4 materials-15-08640-f004:**
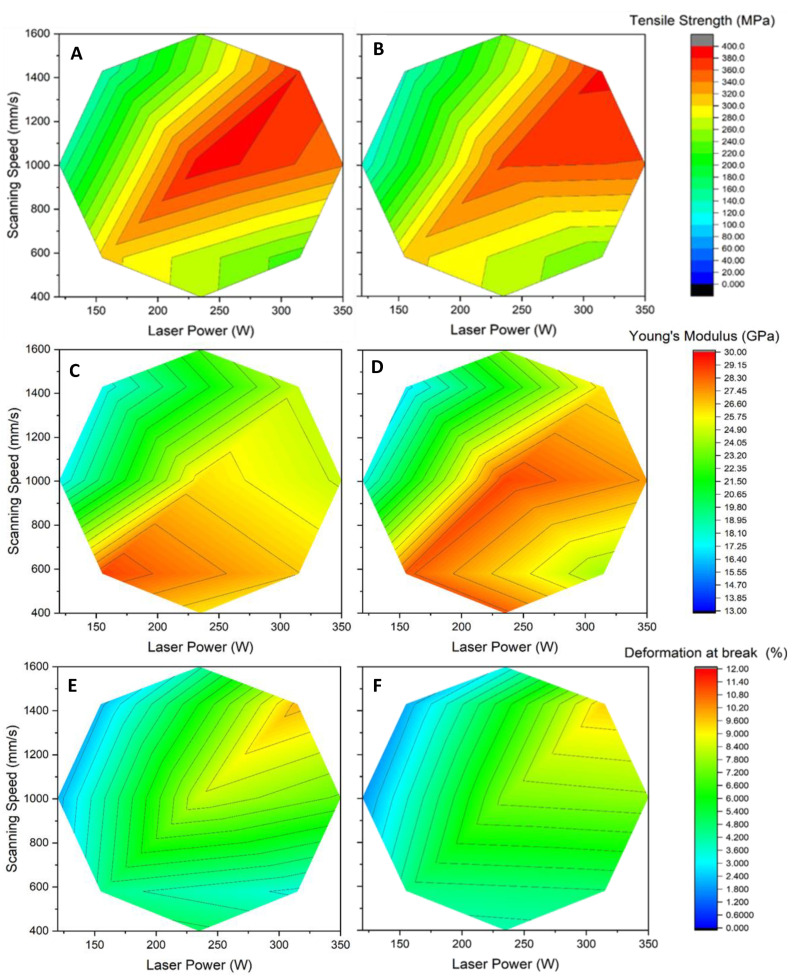
Tensile properties of “as-built” samples as a function of power and speed. Tensile strength using the (**A**) Stripes strategy and (**B**) Chess strategy; Young’s Modulus using the (**C**) Stripes strategy and (**D**) Chess strategy; deformation at break using the (**E**) Stripes strategy and (**F**) Chess strategy.

**Figure 5 materials-15-08640-f005:**
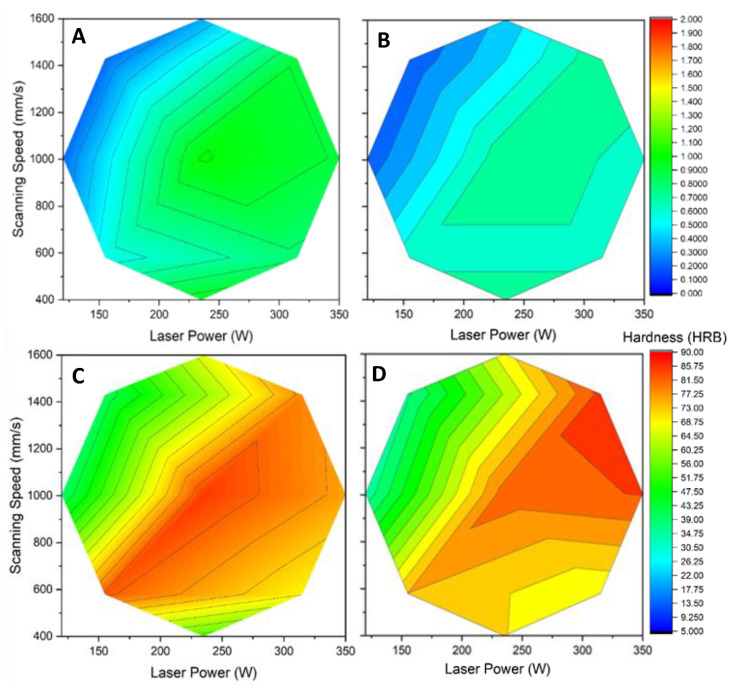
Resilience and hardness of “as-built” samples as a function of power and speed. Toughness (J/cm^2^) using the (**A**) Stripes strategy and (**B**) Chess strategy; hardness using the (**C**) Stripes strategy and (**D**) Chess strategy.

**Figure 6 materials-15-08640-f006:**
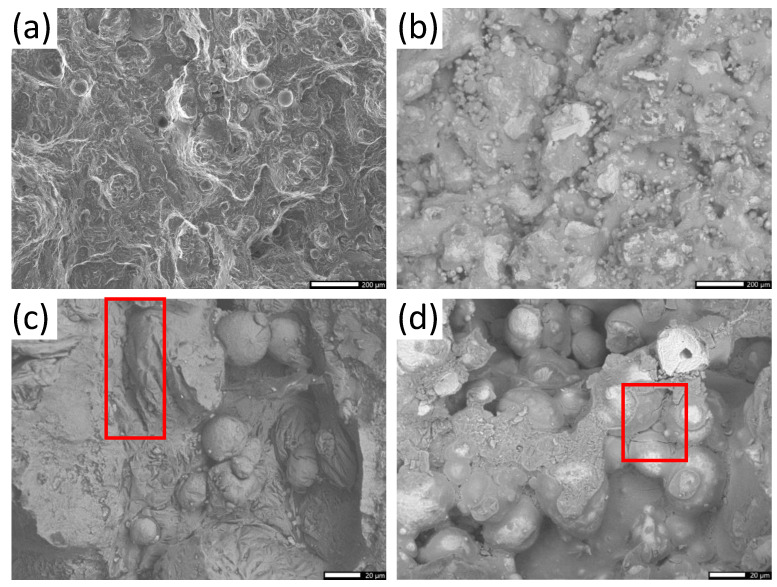
SEM images of the fracture surface of as-built tensile testing samples. (**a**) Test 5 [80X] (**b**) Test 3 [80X] (**c**) Test 5 [600X] (**d**) Test 3 [600X], with the red box highlighting hot cracking.

**Figure 7 materials-15-08640-f007:**
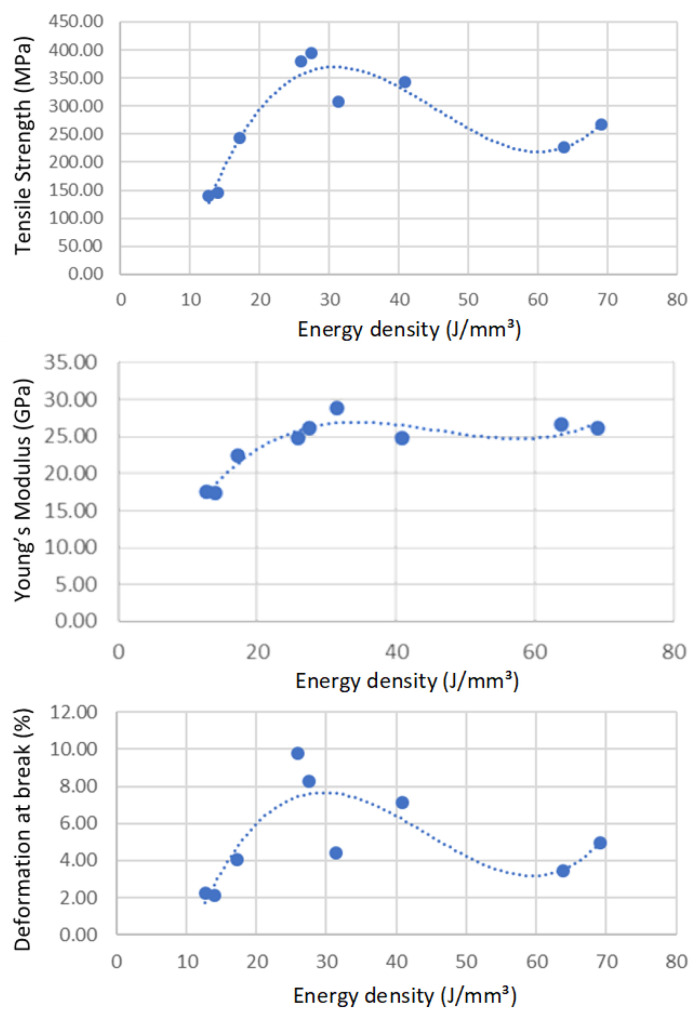
Tensile properties as a function of energy density for as-built parts using the Stripes laser strategy. Top: tensile strength (MPa). Middle: Young’s Modulus (GPa). Bottom: deformation at break (%).

**Figure 8 materials-15-08640-f008:**
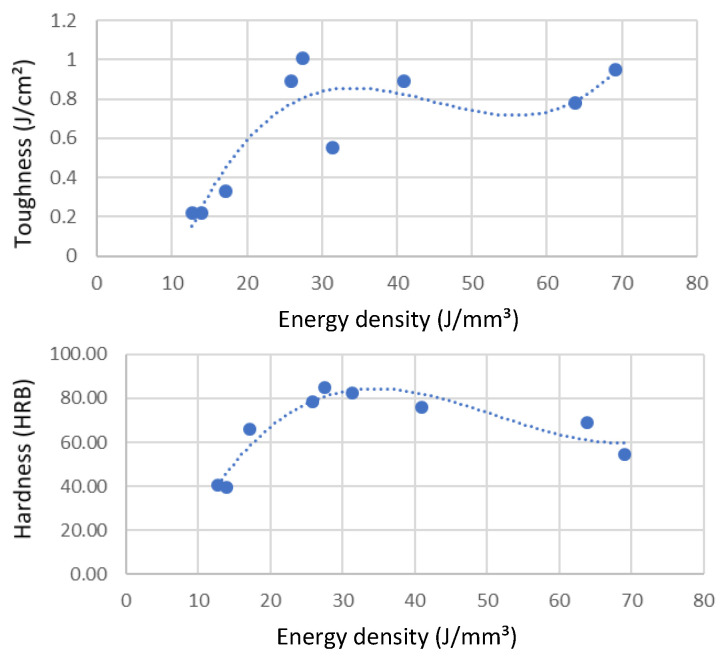
Resilience and hardness of “as-built” samples as a function of energy density, using the Stripes laser strategy. Top: resilience (J/mm^3^). Bottom: hardness (HRB).

**Figure 9 materials-15-08640-f009:**
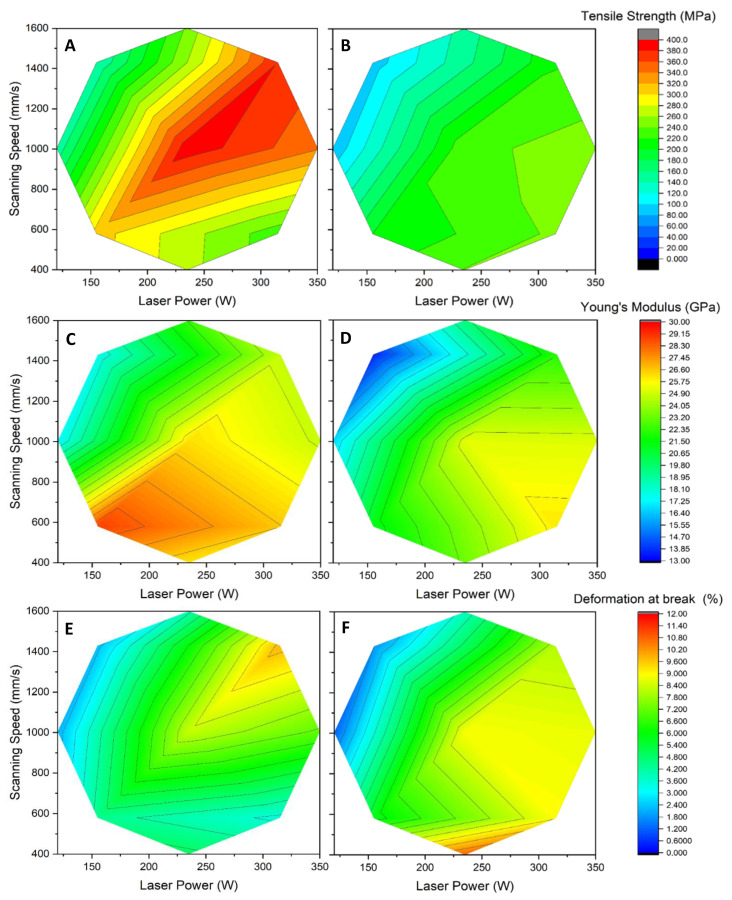
Tensile properties as a function of power and speed. Tensile strength for (**A**) as-built and (**B**) T6 temper; Young’s Modulus for (**C**) as-built and (**D**) T6 temper; deformation at break for (**E**) as-built and (**F**) T6 temper.

**Table 1 materials-15-08640-t001:** Laser parameters used in the composite experimental plan.

Test	Scanning Strategy	Laser Power (W)	Laser Speed (mm/s)	Linear Energy Density (J/mm)	Volumic Energy Density (J/mm^3^)
SLM Solutions	STRIPES	350	1150	0.304	35.8
1	STRIPES	120	1005	0.119	14.0
2	STRIPES	155	580	0.267	31.4
3	STRIPES	155	1430	0.108	12.7
4	STRIPES	235	400	0.587	69.1
5	STRIPES	235	1005	0.234	27.5
6	STRIPES	235	1600	0.147	17.2
7	STRIPES	315	580	0.543	63.8
8	STRIPES	315	1430	0.220	25.9
9	STRIPES	350	1005	0.348	40.9
10	CHESS	120	1005	0.119	14.0
11	CHESS	155	580	0.267	31.4
12	CHESS	155	1430	0.108	12.7
13	CHESS	235	400	0.587	69.1
14	CHESS	235	1005	0.234	27.5
15	CHESS	235	1600	0.147	17.2
16	CHESS	315	580	0.543	63.8
17	CHESS	315	1430	0.220	25.9
18	CHESS	350	1005	0.348	40.9

**Table 2 materials-15-08640-t002:** Regression equations for mechanical properties as a function of energy density.

**STRIPES**	**TENSILE PROPERTIES**		**R^2^**
	Tensile strength (MPa)	$y=0.0119x^{3}-1.6224x^{2}+66.056x-473.38$	0.9135
	Young’s Modulus (GPa)	$y=0.0004x^{3}-0.0502x^{2}+2.1614x-2.8328$	0.8927
	Elongation at break (%)	$y=0.0003x^{3}-0.0453x^{2}+1.7889x-14.392$	0.6830
	**IMPACT**		
	Absorbed energy (J)	$y=0.0001x^{3}-0.0166x^{2}+0.7068x-5.8803$	0.8088
	Tenacity (J/cm^2^)	$y=3E^{-5}x^{3}-0.0038x^{2}+0.1614x-1.3434$	0.7982
	**HARDNESS**		
	Hardness (HRB)	$y=0.0013x^{3}-0.1967x^{2}+9.0177x-44.906$	0.8924
**CHESS**	**TENSILE PROPERTIES**		
	Tensile strength (MPa)	$y=0.0104x^{3}-1.4561x^{2}+61.424x-447.69$	0.9060
	Young’s Modulus (GPa)	$y=0.0006x^{3}-0.081x^{2}+3.2501x-12.778$	0.9623
	Elongation at break (%)	$y=0.0002x^{3}-0.0262x^{2}+1.2078x-9.9125$	0.6885
	**IMPACT**		
	Absorbed energy (J)	$y=0.0001x^{3}-0.0173x^{2}+0.693-5.4657$	0.9514
	Tenacity (J/cm^2^)	$y=3E^{-5}x^{3}-0.0039x^{2}+0.155x-1.22$	0.9498
	**HARDNESS**		
	Hardness (HRB)	$y=0.0018x^{3}-0.2538x^{2}+10.791x-62.304$	0.8880

## Data Availability

Not applicable.

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
