# Peer review of "Effects of Power and Laser Speed on the Mechanical Properties of AlSi7Mg0.6 Manufactured by Laser Powder Bed Fusion"

_materials, 2022, doi:10.3390/ma15238640_

Round 1

Reviewer 1 Report

This paper mainly investigated the effects of laser processing parameters and heattreatment on the mechanical properties of the LPBF-fabricated AlSi7Mg0.6 alloys. Both the topic and the results are lack of novelty. Besides, the provided experimental data lack diversification and the corresponding conclusions are weak. It is obvious that this present version is not ready for publication. The following points need to be considered:

1. The manuscript does not provide sufficient novel knowledge.

2. The studies on the LPBF of AlSi7Mg0.6 alloys are not limited. Many similar investigations can be obtained, including (1) Annales de Chimie-Science Des Materiaux 45 (2021) 1-10 (DOI: 10.18280/acsm.450101); (2) Mater. Des. 109 (2016) 334-346; (3) Inter. J. Adv. Manuf. Technol. 106 (2020) 371-383

3. 71-72 line: keyhole is usually formed at relatively higher laser energy density.

4. 131 line: The diameter of the laser beam spot should be given. How was the platform temperature of 120℃ determined?

5. 222 line: Whether could the authors give the convincing explanation for such large difference in energy density?

6. 293 line: the hot cracks should be marked in Fig. 5 and the label including the equipment information in the SEM images should be removed.

7. 361 line: it is difficult to optimize the laser processing parameters by the fitted equations, in consideration of the low R values.

8. The discussions throughout the whole paper are not profound enough.

Author Response

Reply to reviewer 1 

Author Response

Reply to reviewer 2

Reviewer 3 Report

Dear authors,

Thank you for this very interesting article addressing a good topic. As i am working in this field too, i made different comments along the text. Please, see the reviewed manuscript. I can consider this article for possible publication once all the comments are addressed.

Kind regards

Author Response

Reply to reviewer 3

Round 2

Reviewer 1 Report

The present manuscript has got improved apparently. I think it could be accepted after minor revision. 

1) Why was not the recommended platform temperature of 150℃ used in this study?

2) The texts and the black background (bottom of each image in Fig. 7) should be cut off and the corresponding scales should be re-added.

Author Response

Thank you for your remarks and comments 

Reviewer 2 Report

The revised version is much better than the original manuscript; however, it still needs a major revision, mainly due to the absent evidence supporting some conclusions. The following deficiencies should be addressed:

(i)              The manuscript still requires language editing due to grammar issues. For example, ‘the lasing strategy do not’ (lns. 256-257); ‘it was shown the existence’ (lns. 263-266), ‘cube-textured columnar grains more homogeneous’ (lns. 383-384), among others.

(ii)            As was suggested in my first review, wiser referencing would be good to incorporate. While you addressed this comment by minimizing the citations of proceedings papers in general sentences, some inconsistency still exists. For instance, the paragraph lns. 87-108 addresses the effects of sample orientation and the base plate temperature on the mechanical properties (as suggested in lns. 87-88). However, Rao et al. [20] do not consider these effects. Furthermore, you mention the orientation effect in lns. 87-88 but never discussed it.

(iii)          In my opinion, the novelty became better formulated; however, it still requires further editing. Please compare your research against the published literature on the effects of energy density and scanning strategy on the mechanical properties of AlSi7Mg0.6 alloy and highlight what your study brings to the community (thicker layers, specific scanning strategies, etc.).

(iv)           Please comment on why a lower base plate heating temperature was chosen as compared to the recommended one (lns. 142-145).

(v)             I would recommend clearly mentioning in the Materials and Methods section how many samples you used for which tests. You began to do it but spread it into two sections (Materials and Methods and Results) for some reason.

(vi)           Please provide the formula for the linear energy density as well (lns.234-235).

(vii)         Please provide the evidence for the microstructure described (lns. 342-350) or omit this discussion. The same holds true for Young’s modulus and deformation at break values for the chess scanning strategy (lns. 399-400) and after heat treatment (lns. 469-471). You could always create an appendix if you do not want to overload the paper. However, I do not see much pain in adding those for the chess strategy to the existing figures for the stripes scanning.

(viii)       As mentioned in the first review, evidence should be provided to support the conclusion in lns. 518-520. The manuscript should discuss this study result, not something upcoming. If there is no result, there should be no conclusion about it. The same applies to the scanning strategy effects on the mechanical properties/thermal management (lns. 511-513). First, please specify the scanning strategies in the conclusion. Secondly, there is no evidence in the manuscript demonstrating that the scanning strategy affects thermal management more than mechanical properties.

Author Response

Thank you for your remarks and comments 

Reviewer 3 Report

Authors have improved the manuscript after all the considerations from reviewers. Therefore, this can be considered ready for publication. However, as authors replied in one comment, we will have to wait for an adittional publication to see the desired microstructures!

Author Response

Thank you for your remarks and comments 
